# Visual Cognition and the Science of Magic

**Geoff G. Cole [1],\* and Abbie C. Millett [2]**

1   Centre for Brain Science, University of Essex, Wivenhoe Park, Colchester CO4 3SQ, UK
2   School of Social Sciences and Humanities, University of Suffolk, Ipswich IP4 1QJ, UK; a.millett2@uos.ac.uk
\*   Correspondence: ggcole@essex.ac.uk

**Abstract:** A number of authors have argued that the art of conjuring can assist the development of theories and knowledge in visual cognition and psychology more broadly. A central assumption of the so-called science of magic is that magicians possess particular insight into human cognition. In a series of experiments, we tested the Insight hypothesis by assessing three factors that magicians argue are important for a popular illusion. Participants viewed videos of a magician performing the French Drop sleight whilst gaze, motion, and muscular tension were manipulated across experiments. Contrary to what the community of conjurers state, results showed that none of these influenced the perceived success of the effect. We also found that a visual priming technique, one suggested of many and used by an eminent magician, does not influence participant responses. Overall, these findings fail to support the Insight hypothesis. We suggest that scientists of magic have erroneously imbued magicians with insights they do not possess.

**Keywords:** science of magic; priming; gaze cueing; motion capture; Derren Brown; Teller





## 1. Introduction

The science of magic has seen enormous growth since the inception of its modern form around 15 years ago. Over 150 papers have now appeared, covering a wide range of topics such as problem solving [1], well-being [2], agency [3], motion [4], belief formation [5], artificial intelligence [6], eyewitness testimony [7], expertise [8], motor rehabilitation [9], amodal completion [10], and autism [11]. These are all in addition to the work on visual attention (see [12] for an early review), the field that have been central to the science of magic endeavour.

A major argument presented by the proponents of the movement is that magicians, with their collective vast experience of manipulating perception, possess particular insight and knowledge of human behaviour. For example, Otero-Millan, Macknik, Robbins, McCamy, and Martinez-Conde [13] wrote that "the magician's intuitive understanding of the spectator's mindset can surpass that of the cognitive scientist" (p. 1). Similarly, Martinez-Conde and Macknik [14] argued that magicians possess a "deep intuitive understanding of the neural processes taking place in the spectators' brains" (p. 72) and that "Neuroscientists are just beginning to catch up" (p. 74). At the very least, magicians are said to have "intuitive knowledge about the rules governing human cognition" ([15]; p. 117). However the hypothesis is described, and whatever one feels about the (above) phrases used to illustrate the point, the Insight notion has been central to the science of magic endeavour. Furthermore, some of the effects that magicians have developed, many of which predate their assessment by psychologists, have also been taken as evidence for the Insight hypothesis. One of the most well-known examples is 'misdirection', e.g., [16–18], in which a spectator's attention is directed from the location of a secret method. Perhaps beginning with the work of Michael Posner in the 1980s, e.g., [19], the marshalling of attention, or "attentional capture", became a major subfield within cognitive psychology.

The insight argument is supported by Pailhès and Kuhn [20], who adopted a proposal made by the eminent British illusionist Derren Brown [21]. The authors tested Brown's

suggestion that a spectator can be primed to choose a particular playing card by use of subtle verbal cues and hand gestures made by the magician. Participants in the experimental group were presented with a video (or a live interacting magician) in which she asked the participant to think of a "bright and vivid" card (thus attempting to prime a red suit) whilst also subtly forming a diamond shape then performing a movement that mimicked the number three, all with the hands. The effect was confirmed; participants were more likely to choose the three of diamonds when primed compared to when no prime had occurred.

A similar insight effect was reported by Otero-Millan et al. [13]. The authors collaborated with the magician Apollo Robbins, who suggested that spectator eye gaze could be more effectively directed away from the secret location of a small hidden object by a curved and somewhat exaggerated motion of the magician's hand. In the French Drop sleight, the magician appears to transfer a coin (or any small object) from one hand to the other when the coin is in fact retained at the original (i.e., hand) location. This can be used to generate an illusion in which the item seems to disappear. Otero-Millan et al. found that a participant's eye gaze was indeed more likely to shift from the secret action by a curved hand movement compared to when no such motion occurred.

Cole [22] has however argued that science of magic researchers should be cautious in developing methods based on magician 'insight'. He reminds us that magicians are primarily concerned with entertainment and secrecy. Consequently, one can never be sure if a magic effect is based on the principle that the psychologist thinks it is. Cole noted the developing trend in which many magicians falsely state that they are using a subtle psychological principle (e.g., attentional manipulation; subliminal suggestion) when, in, fact they are using a classic magic procedure: a procedure that has no interesting psychological component (e.g., hidden compartments). Furthermore, even when magicians are not consciously making false claims, they are of course susceptible to the effect in which humans often believe they have more control over events in their immediate environment than they actually do [23]. Magicians may therefore be *led to believe* that a particular technique they employ influences spectator behaviour.

In a series of eight experiments, we tested the insight hypothesis. That is, we tested the claim that conjurers possess particular knowledge of visual cognition pertaining to illusions they commonly perform. In the first four experiments, we examined a common conjuring belief concerning success of the French Drop sleight. Experiments 1 and 2 tested the influence of the magician's gaze, and Experiments 3 and 4 examined the influence of hand motion. Experiments 5 and 6 examined a hypothesis put forward by the renowned magician *Teller*, again concerning the French Drop procedure. Experiments 7 and 8 tested the Derren Brown notion that the inclusion of the syllable 'King' into a 'pick a card' instruction will prime a participant to choose a King playing card.

## 2. Gaze and the French Drop

### 2.1. Experiment 1

Magicians are experienced at manipulating an observer's attention. Although not all "misdirection" is spatial, it is often necessary to shift an audience's attention away from the location of a secret move for an illusion to be effective. One standard suggestion is that the magician should use their own gaze direction to achieve this. As Ortiz [24] noted, spectators "look where you look". Historically, this principle has always been applied to the French Drop sleight described above. Indeed, the belief in the importance of gaze in this effect has accrued axiom status amongst magicians. For instance, the classic text Modern Coin Magic [25] states that the magician should keep their attention fixed on the hand that appears to take the coin ("Keep your eyes on the closed left hand"). In other words, away from the hand where the coin is actually located. The clear implication is that the effect has less chance of succeeding if this advice is not adopted.

In our first experiment, we tested this assumption. Participants were presented with one of two short videos of a magician performing the standard French Drop (see Figure 1). In one condition, the magician's face, and thus eye gaze, could be seen whist in the other

his whole head was obscured. Participants were then asked to indicate where at the end of the sequence they thought the coin was located. This can therefore be seen as a measure of how successful the method is. If the magician's gaze contributes to the effect, success of the sleight (i.e., non-detection) should be greater in the Gaze condition.

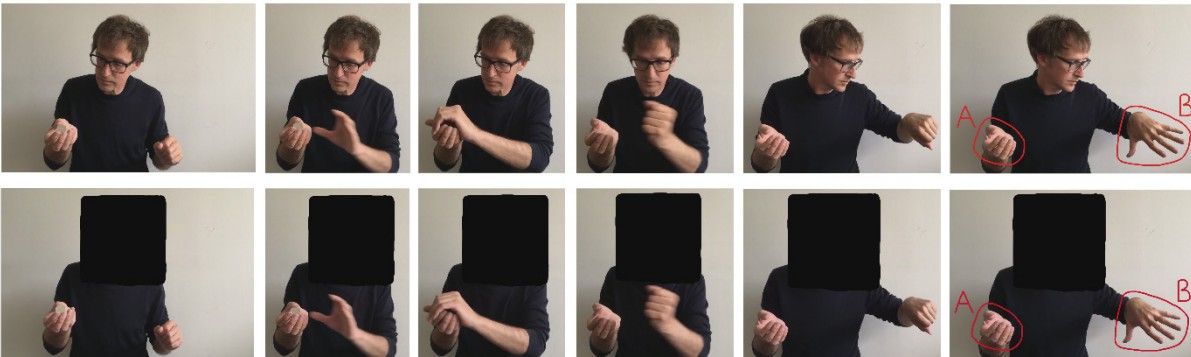

**Figure 1.** Stills from the videos presented in Experiment 1. The upper panels show the sequence pertaining to the Gaze condition and the lower pertaining to the No Gaze condition. The Gaze condition is a typical procedure used in the French Drop sleight. Here, the magician's left hand moves across to the right hand and appears to take the coin. The left hand in fact obscures the secret move, i.e., the dropping of the coin from the tips of the fingers (of the right hand) to the palm (of the right hand).

### 2.1.1. Method

Participants

There were 96 participants (74 m, 22 f). The mean age was 29.7 years (*SD* = 7.2). Note that with the exception of Experiment 8, a different sample of participants was recruited for each experiment.

Stimuli and Procedure

The stimuli and sequence of events can be seen in Figure 1. A male was filmed performing a standard French Drop in which a coin was retained in the right hand. The model gazed at the left hand as it moved away from where the coin swap looked to have occurred. At the end of the sequence, two circles appeared overlaid onto each of the magician's hands. One was labelled "A", and the other was labelled "B". The film lasted for 10 s and constituted the "Gaze" condition. For the "No Gaze" condition, the model's head was obscured with a black square box. Participants were randomly allocated to one of the two conditions, but it was ensured that there would be 48 participants in each condition. The experiment was administered using the online platform Qualtrics, and participants were compensated financially for their time using Amazon Mechanical Turk. After seeing an initial display in which general information about the study was presented, participants were given the following instructions. "You will see a 10 s video in which a magician manipulates a coin. Your task is to decide whether the coin ends up at Location "A" or Location "B"". Participants thus provided a single response. The experimental set up did not allow participants to see the video more than once. After responding, participants then undertook the present Experiment 8. The research received ethical approval from the University of Essex Psychology Ethics Committee for all eight experiments (Approval reference GC1903) on 23 July 2019. All videos are available at https://drive.google.com/drive/folders/19wyub8qJucqt1DcBEi860jIjuAtfqYm5?usp=sharing (accessed on 1 July 2023). The data were collected during 2019.

### 2.1.2. Results

In the Gaze condition, 9 of the 48 participants responded that the coin was located at the gazed-at location (i.e., Location "B"). The same proportions also occurred in the "No

Gaze" condition, i.e., 9 of the 48. A Chi-square test of independence would therefore be 0 and the *p* value would be 1. In all the present experiments, we employed Bayesian analyses to confirm the effects observed with Chi-square tests. Unlike traditional inferential statistics, Bayesian analyses reveal how likely one event (e.g., a difference between conditions) is relative to another event (e.g., no difference between conditions). For example, a Bayes factor of 4 shows that one event is four times more likely than the alternative. Similarly, a Bayes factor of 0.25 shows that this same event is four times less likely than the alternative. A Bayesian analysis, using the code described by Dienes, e.g., [26], showed that the null hypothesis was more than twice as likely than the alternative, BF = 0.46. These data do not therefore support the assumption that the magician's gaze contributes to the French Drop effect.

### 2.2. Experiment 2

Faces have been shown to attract attention using a number of paradigms. For instance, reaction times to locate a target in a display are increased when a face is also present [27]. Similarly, Ro, Russel, and Lavie [28] found that faces are less susceptible to change blindness compared with non-face objects (e.g., clothes). It is therefore possible that the obscuring of the model's face and head in Experiment 1 had some unknown and unintended effect on responses. It is perhaps inevitable that participant attention was more focused on the lower portion of the video (i.e., where the effect occurred) in the "No Gaze" condition, because a large section of the upper portion was obscured with a black square. In Experiment 2, we replicated the procedure of Experiment 1 with the sole exception that new stimuli were generated. In one video (i.e., the "No Gaze Misdirecting" condition), the magician gazed at the hand that retained the coin. In the other video ("Gaze Misdirecting"), the magician's gaze followed the hand that did not retain the coin. The latter condition is the standard procedure in which gaze is used to shift attention away from the coin's real location. If the magician's gaze contributes to the French Drop sleight, the success of the effect should be greater in the Gaze Misdirecting condition.

### 2.2.1. Method

All aspects of the method were as described for Experiment 1 with the following exceptions. An additional independent sample of 76 males and 20 females whose average age was 29.2 years (*SD* = 7.8) were recruited. The only other difference were the stimuli. The sequence of the two videos is shown in Figure 2.

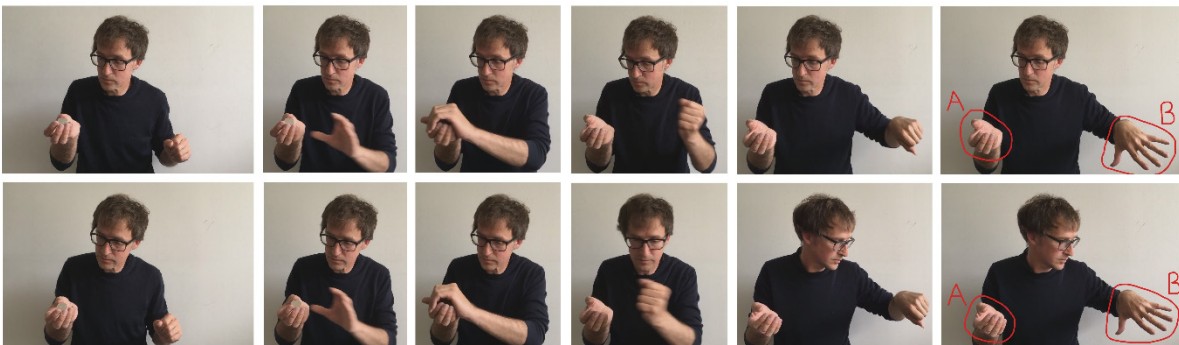

**Figure 2.** Stills from the stimuli presented in Experiment 2. The upper panels show the No Gaze Misdirecting condition and the lower show the Gaze Misdirecting.

### 2.2.2. Results

In the Gaze Misdirecting condition, 12 of the 48 participants responded that the coin was located at the gazed-at location. In the No Gaze Misdirecting condition, 14 of the 48 did so. A Chi-square test of independence found no significant effect, $X^2$ (1, N = 96) = 0.21, *p* > 0.64. Bayesian analysis showed that the null hypothesis was two and a half times

more likely than the alternative, BF = 0.4. These results thus concur with the findings from Experiment 1. The French Drop sleight is not influenced by the magician's gaze location.

## 3. Motion and the French Drop

### 3.1. Experiment 3

In addition to gaze direction, motion is also argued to play an important role in the French Drop. Indeed, this was a central motivation of the Otero-Millan et al. [13] research described in the present Introduction. As with gaze, the rationale is that the movement of one hand will shift attention away from where the coin is retained. Although this idea predates experimental psychology, research on motion capture seems to support this assertion. It is well known that salient events such as movement onset are particularly effective at shifting attention. For example, Abrams and Christ [29] presented participants with displays of items, one of which became a target. Although irrelevant to the task, one item was constantly moving, another item began to move, a third item stopped moving, and a fourth never moved (i.e., static). Results showed that search for the target was most efficient when the target happened to coincide with the item that began to move. This suggests that the onset of movement is particularly effective at attracting attention.

In Experiment 3, we tested the French Drop motion hypothesis. During the typical procedure, the magician will use both their gaze and motion to shift attention away from the location of the coin. We therefore isolated the effect of motion by not showing any part of the model's head and face. Participants again viewed one of two videos. In one condition ("Motion Equated"), the degree of movement of the two hands was the same. Thus, the motion signal was equated with respect to the coin's location. (Virtually) the only motion that was present in the other condition ("Motion Misdirection") aimed to draw attention away from the location of the coin. This is in effect the standard procedure used by many illusionists. If motion of the hands is indeed important in the French Drop sleight, success of the effect should be greater in the Motion Misdirection condition.

#### 3.1.1. Method

All aspects of the method were as described for Experiment 1 with the following exceptions. An additional independent sample of 60 males, 35 females and 1 participant who indicated that they preferred not to state their sex were recruited. The mean age was 36.5 years (*SD* = 10.5). The only other difference was the stimuli. The sequence of the two videos is shown in Figure 3. In the Motion Equated condition, the magician began the sequence with each hand to the side of the body. They were then drawn together, and the Drop occurred at the midpoint in their trajectory. In the Motion Misdirection condition, (virtually) all the movement was generated by the trajectory of one hand: the hand that did not possess the coin.

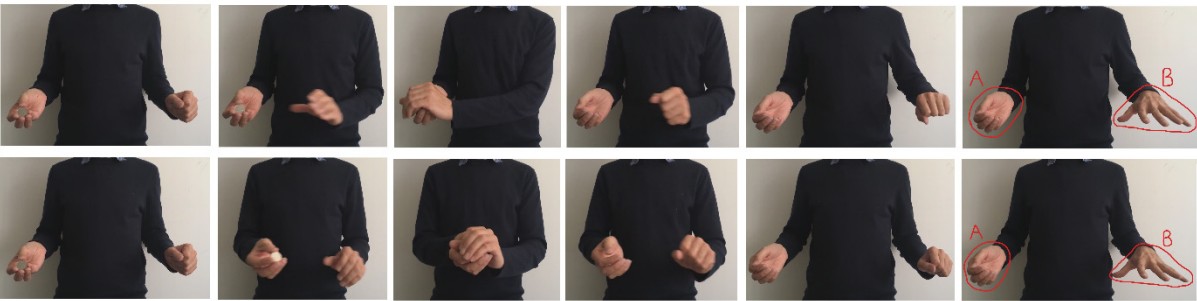

**Figure 3.** Stimulus sequence in Experiment 3. The upper panels show the Motion Misdirection condition and the lower show the Motion Equated condition. In the former condition, only the left hand moves, i.e., the hand that appears to take the coin. In the latter condition, an equal amount of movement occurs in both hands.

### 3.1.2. Results

In the Motion Misdirecting condition, 11 of the 48 participants responded that the coin was located in the moving hand. In the Motion Equated condition, 9 of the 48 did so. This small difference was not significant, $X^2$ (1, N = 96) = 0.25, $p$ > 0.61. A Bayesian analysis showed that the French Drop motion hypothesis is less likely than the null hypothesis (BF = 0.74).

### 3.2. *Experiment 4*

Rather than comparing equated motion with (presumed) misdirecting motion, Experiment 4 compared the success of the French Drop when all the motion occurred at the hand that retained the coin; alternatively, it all occurred at the hand that was stationary (see Figure 4). The motion hypothesis again predicts that the effectiveness of the French Drop should depend on the location of the motion. Specifically, detection should be reduced in the Coin at Motion hand condition.

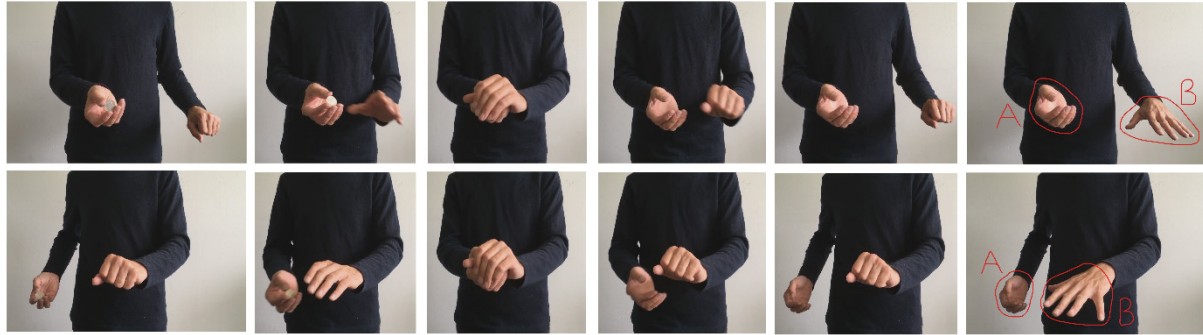

**Figure 4.** Stimulus sequence in Experiment 4. The upper panels show the Coin at Stationary Hand condition and the lower show the Coin at Motion Hand condition. In the former condition, only the left hand moves, i.e., the hand that appears to take the coin. In the latter, only the right moves. Note that this condition contravenes the central principle of how hand motion should be used in the French Drop; all the (attention capturing) motion is potentially shifting attention to the secret location of the coin.

### 3.2.1. Method

All aspects of the method were as described for Experiment 1 with the following exceptions. An additional independent sample of 78 males and 18 females whose mean age was 29.7 years (*SD* = 7.6) were recruited. The sequence of the two videos is shown in Figure 4. In both, the magician began the sequence with one hand to the side of the body and the other at the midline. The coin was always located in the right hand. In the Coin at Motion Hand condition, he moved his right hand across to the left and performed the Drop; then, he moved his right hand (still containing the coin) back to its starting position. In the Coin at Stationary Hand, he moved his left hand across to his right, performed the Drop, then moved the (empty) left hand back to its starting position.

### 3.2.2. Results

In the Coin at Motion Hand condition, 12 of the 48 participants responded that the coin was located in the moving hand. In the Coin at Stationary Hand condition, this was 13 of the 48. This small difference was not significant, $X^2$ (1, N = 96) = 0.05, $p$ > 0.81. A Bayesian analysis also found no evidence for the hypothesis that motion contributes to success of the French Drop, i.e., the null hypothesis is more than twice as likely as the alternative (BF = 0.46).

## 4. Muscular Tension and the French Drop

### 4.1. Experiment 5

Phillips, Natter, and Egan [30] reported a further principle that magicians are said to use in order to aid the effectiveness of the French Drop. In a personal communication to Phillips et al., the eminent magician Teller described how the muscular tension of the hand holding the coin is particularly rigid compared to the free hand. This, according to Teller, can be exploited by making the hand that appears to take the coin similarly tense at the point of the apparent switch. In other words, there is a transfer of tension from one hand to the other.

In Experiment 5 we tested the Muscular Tension hypothesis. As previously, participants saw one of two videos. In one, the magician clasped the free (i.e., empty) hand immediately after the switch (supposedly) took place and then pantomimed a snatching motion of that hand to further emphasise the switch (the "Muscular Tension" condition). In the other video/condition ("No Muscular Tension"), no such clasping and motion occurred. Although the present experiment tested the Muscular Tension hypothesis, it can also be seen as a further assessment of the motion hypothesis examined in Experiments 3 and 4. Indeed, the movement of the empty hand in the Muscular Tension condition is very similar to the curved motion condition of Otero-Millan et al. [13].

#### 4.1.1. Method

All aspects of the method were as described for Experiment 1 with the following exceptions. An additional independent sample of 60 males and 36 females whose mean age was 39.6 years (*SD* = 9.8) were recruited. The stimuli are shown in Figure 5. In both, the magician held the coin in his right hand where it was retained after the Drop. As described above, in the Muscular Tension condition, he exaggerated the gripping and movement of the left hand immediately after the Drop. This contrasted the No Muscular Tension condition in which no such movement occurred.

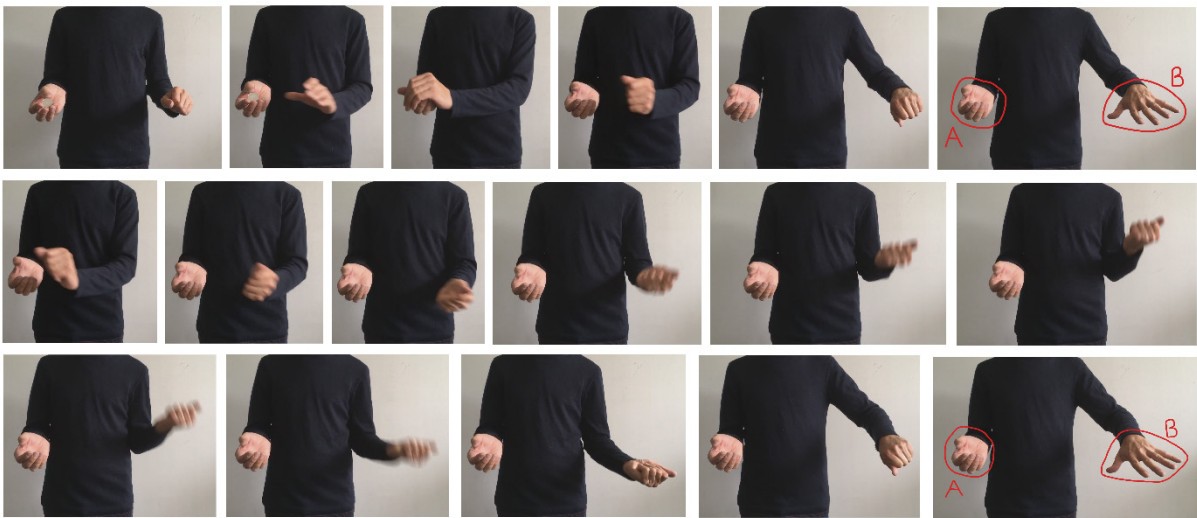

**Figure 5.** Stimulus sequence in Experiment 5. The upper row shows the No Muscular Tension condition. The lower two rows show the Muscular Tension condition (i.e., from the middle left to bottom right). Note that the sequence shown here for the Muscular Tension condition begins immediately after the Drop has occurred.

#### 4.1.2. Results

In the Muscular Tension condition, 3 of the 48 participants indicated that the coin was located in the tensioned hand. This contrasted 9 out of 48 in the No Muscular Tension condition. This difference was not significant, $X^2$ (1, N = 96) = 3.4, *p* = 0.06. The lack of evidence for the experimental hypothesis was confirmed with a Bayesian analysis. The

null hypothesis was almost four times more likely than the alternative, BF = 0.27. Note that the small BF, in contrast to the almost significant Chi-square value, is because the pattern of data is in the opposite direction to that predicted by the muscular tension hypothesis. Overall, these results suggest that an increase in muscular tension immediately after the switch during the French Drop does not contribute to the effect.

### 4.2. Experiment 6

One intriguing aspect of the results from Experiments 1 to 5 is the low proportion of responses indicating that the French Drop effect was successful as an illusion. Of all the 10 conditions undertaken, only 21% of participants responded that the coin was located in the empty hand. The French Drop seemed to be similarly unsuccessful (just above chance, 55.9%) when performed by an expert magician in the Phillips et al. [30] study. In the present experiments, the success rate was clearly below chance. One possible reason is that the magician was always seen opening the empty hand (i.e., Hand B) at the end of the video. He also kept the coin-retained hand (i.e., Hand A) face-up. This may have reduced the likelihood of the coin being deemed to be located at the empty hand. Indeed, we deliberately kept Hand B face-down and Hand A face-up in order to induce uncertainty and therefore reduce the success of the illusion. We were conscious of how powerful the French Drop is and thought that there would be a ceiling effect with a high proportion of participants presuming the coin had been swapped in both conditions. As the data show, this aspect of our stimuli was more successful than we anticipated; only 21% of participants were susceptible to the illusion.

Although the critical analysis in Experiments 1–5 was the comparison between conditions (rather than overall success), in Experiment 6, the magician performed the same muscular tension used in Experiment 5 (in the Muscular Tension condition) but did not open the empty hand. In effect, there was no illusion for the participant to consider; they simply saw the magician pass a coin from one hand to the other.

#### 4.2.1. Method

The method was as described previously with the following exceptions. An additional independent sample of 57 males and 39 females were recruited, with a mean age of 38.6 years (*SD* = 10.7). The stimuli were identical to those used in Experiment 5 with the sole exception that the video ended immediately before the magician opened his empty left hand. At this point, the "A" and "B" circles appeared.

#### 4.2.2. Results

In the Muscular Tension condition, 11 of the 48 participants indicated that the coin was located in the tensioned hand. This figure was 13 out of 48 in the No Muscular Tension condition. This difference was not significant, $X^2$ (1, N = 96) = 0.22, $p > 0.63$. The failure to support the experimental hypothesis was also found with a Bayesian analysis (BF = 0.4), showing that the null hypothesis is two and a half times more likely than the alternative hypothesis. These data thus concur with those of Experiment 5; the perception of muscular tension does not contribute to the French Drop effect. The results also show that although keeping the empty hand closed at the end of the routine did significantly increase the success rate of the illusion (i.e., 24 out of 96 participants versus 12 out of 96 in Experiment 5; $X^2$ (1, N = 96) = 4.9, $p = 0.03$.), this increase did not modulate any effect of muscular tension on the French Drop slight. There does however remain the possibility that if Hand A was positioned palm-down and Hand B remained closed, this could generate an effect of the central manipulation (e.g., muscular tension).

## 5. Priming a King

### 5.1. Experiment 7

Recall from the present Introduction that one of Derren Brown's priming procedures (or "forces") was empirically supported by Pailhès and Kuhn [31]. In a review of the science

of magic, Thomas et al. [15] referenced a further psychological technique (supposedly) used by Brown. Thomas et al. stated that "to force a card, the magician might talk to spectators about "making a mental picture of the card. This phrase could lead to the choice of a king, because the syllable [king] in the word making acts as a prime". Thomas et al. also added that Brown's psychological methods, "seem to involve a still poorly understood mental-influence technique based on a new type of priming, one that is more subtle and more implicit than those generally used in experimental psychology".

In Experiment 7, we tested Brown's "king" priming hypothesis. Participants were presented with a video in which an adult male either asks them to "start thinking" of a playing card or asked them to "choose" a playing card.

### 5.1.1. Method

All aspects of the method were as described previously with the following exceptions. An additional independent sample of 103 participants; 17 males, 85 females, and one non-binary participant were recruited using SONA, the University of Suffolk's online participation pool. All were first year psychology undergraduates who, at the time of testing, had not received any degree-level tuition on attention. The mean age was 26.3 years (*SD* = 7.79). Participants were randomly assigned to one of the two conditions (but without the stipulation that there should be the same number in each condition). This resulted in 47 participants being assigned to the priming condition and 56 to the non-priming condition. In the priming condition, the man looked directly into the camera and stated, "Start thinking of a random high value playing card, higher than 9. What card did you think of"? In the non-priming condition he stated, "Chose a random high value playing card, higher than 9. What card did you choose"? They were then presented with a drop down menu showing the Ten, Jack, Queen, King, and Ace for all four suits.

### 5.1.2. Results

In the non-priming condition, 12 of the 56 participants chose a King with the rest (44) choosing a non-king. In the priming condition, 13 of the 47 participants chose a King and the rest (34) choose a non-king. There was no significant effect of the prime, $X^2$ (1, N = 103) = 0.54, $p > 0.46$. The null hypothesis was also more likely than the experimental hypothesis, BF = 0.8. These results do not therefore support the *King* priming hypothesis.

### *5.2. Experiment 8*

It is possible that any King effect only occurs when participants are presented with the prime as written text. Therefore, in Experiment 8, participants read the instructions as opposed to hearing them.

### 5.2.1. Method

All aspects of the method were as described previously with the following exceptions. We vastly increased Power by recruiting 600 participants, 424 males, and 175 females. One participant did not indicate their sex. As stated earlier, all the participants took part in Experiment 1–6. We then added 24 additional participants. The mean age was 33.9 years (*SD* = 9). Participants were randomly assigned to either the priming condition or the non-priming condition. A total of 294 participants became assigned to the former and 306 to the latter. In the priming condition, participants were given the following text "Start thinking of a random high value playing card, higher than 9. What card did you think of"? In the latter, they were given, "Choose a random high value playing card, higher than 9. What card did you choose"?

### 5.2.2. Results

In the non-priming condition, 59 of the 306 participants chose a King with the rest (247) choosing a non-king. In the priming condition, 61 of the 294 participants chose a King and the rest (233) chose a non-king. There was no significant effect of the prime, $X^2$

(1, N = 600) = 0.2, *p* > 0.65. This was confirmed with a Bayesian analysis, BF = 0.35. These data show that the inclusion of "king" within written text does not prime a participant to choose a *King* playing card. As with Experiment 7, these findings do not support the King priming hypothesis.

We finally performed a Bayesian analysis on all responses in all of our eight experiments. In effect, this meant that we tested the hypothesis that magicians possess insight into how the French Drop works as an illusion (i.e., via gaze, motion, and muscular tension), as well as a possible priming phenomenon. The null hypothesis is that no such insight occurs. We found that the null hypothesis was almost seven times more likely than the Insight hypothesis (BF = 0.15).

## 6. General Discussion

The present series of experiments examined the hypothesis that magicians possess particular insight into human cognition pertaining to illusions. We examined whether a magician's gaze and hand motion are important for success of the French Drop. Contrary to popular belief within the profession, gaze and motion do not influence this classic sleight and neither does the pantomiming of muscular tension in the hand that appears to take the coin. We have also found that a *King* playing card is not primed by the use of the syllable "king" in the instructions. These data do not therefore support the Insight hypothesis. It is of course possible that the suggestions we have tested just happen to be ones that do not influence participant responses. We did however test at least two central assumptions of one of the oldest procedures: that is, the importance of gaze and motion in the French Drop sleight. Indeed, as noted in the Introduction, the importance of gaze has received axiom status within conjuring. We have also tested suggestions made by two of the most eminent magicians of the past two decades.

Rather than possessing particular insight into human cognition and behaviour, we suggest that magicians are subject to the phenomenon briefly referred to in the Introduction. Humans tend to believe they have greater control of events than they actually do [23]. One can imagine this is particularly the case for magicians who *do* have a large amount of control of the performance environment. This may well have led illusionists to erroneously believe they have more control than is actually the case. Although magicians do manipulate spectator attention during the French Drop [13], they have mistakenly thought that attention is an important component in the success of the procedure. The present data instead support the findings from one of the few previous studies to assess magicians' knowledge of the mechanism that underlie an illusion. Although this aspect of the work was not of principal interest, Pailhès and Kuhn [31] asked 91 magicians to rate the importance of eight different factors that contribute to the success of the so-called Criss-Cross force. A time delay that is typically included in the procedure and an attentional misdirection were considered to be the two most important factors. However, when Pailhès and Kuhn experimentally manipulated these, they found that neither contribute to the Criss-Cross force.

The present results support the view [22] that the Pailhès and Kuhn [20] priming-a-card finding (see present Introduction) was a false positive. Although that particular Derren Brown suggestion is a real effect, it is one of dozens of psychological techniques that he has posited in a number of popular magic books and television shows. For instance, Brown [21] can persuade a small group of people to rob a cash delivery van by presenting them with a number of subtle primes over a number of days, e.g., displaying the acronym "KASH" via PowerPoint during a seminar. Although this makes for great television, problems arise when experimental psychologists attempt to adopt and assess some of these techniques. Essentially, the cognitive scientist can never be sure which effects have an interesting psychological basis (e.g., priming) and which do not. This is further compounded by the fact that some stage magicians will now state that they are about to employ a psychological principle when no such principle is applied, e.g., [32,33]. This contrasts the illusion of control principle mentioned above, in which the magician at least believes he/she is using psychology.

Although we have challenged the Insight hypothesis, our concern is not with magicians or any of the claims they make. They are after all entertainers, and there is no detriment to employing techniques that they (erroneously) believe contribute to a given effect. Our concern is with the scientists of magic who have imbued magicians, as a collective group, with insight they do not possess. Relatedly, the science of magic has exaggerated the knowledge that can be gained from studying magic tricks. The French Drop is a case in point. The effect is a clearly a perceptual rather than an attentional phenomenon. Indeed, it has no attentional component. In cognitive psychology, attentional limitations typically refer to effects in which an observer fails to detect an event despite it being fully visible. A perceptual effect in contrast refers to an observer's ability to actually see the critical stimulus. If, in a control condition, an event goes undetected even when a participant's attention is directed to the location of the event, the phenomenon is deemed not to be attentional in nature. This is usually because the event is not perceptible; it cannot be seen. This is the situation we have with the French Drop sleight. The central reason why the Drop is effective is because the critical event (i.e., the coin dropping into the palm) is obscured by the hand that appears to take the coin. If this occlusion did not occur, there would be no effect, unless attention was used to generate the effect. To put this another way, if a participant was told to attend to the critical region (as they naturally do in the course of the sleight), they would still not see the drop. This can be contrasted with another illusion used in research [17] in which a cigarette was dropped from the magician's hand. Although this went unnoticed by the majority of participants, it was fully visible; it was an attentional effect. This explains why detection rate vastly increased when Kuhn et al. showed participants the illusion for the second time. It is therefore inevitable that an attention manipulation in the French Drop will fail to modulate detectability when attention is not involved in the illusion.

In sum, the present results suggest that magicians are unlikely to have particular insight into human cognition and behaviour. Principles that have historically been considered important for success on the classic French Drop illusion (i.e., gaze location and motion capture) and a different principle suggested by an eminent magician (i.e., muscular tension) do not contribute to that effect. We have also found that word priming does not influence the choice of a playing card, as suggested by another eminent magician. We suggest that the science of magic has erroneously imbued magicians with insight they do not possess.

**Author Contributions:** G.G.C. conceived of the work, generated the stimuli, analysed the data and wrote the manuscript. A.C.M. organised data collection and edited the manuscript. All authors have read and agreed to the published version of the manuscript.

**Funding:** The present work was funded by the University of Essex. There are no relevant financial or non-financial interests to disclose.

**Institutional Review Board Statement:** The study was conducted in accordance with the British Psychological Society code of Ethics, and approved by the University of Essex Ethics Committee (Approval reference GC1903 on 23 July 2019).

**Informed Consent Statement:** Informed consent was obtained from all subjects involved in the study.

**Data Availability Statement:** The datasets pertaining to all our experiments are available at https://osf.io/jnz7t/ (accessed on 1 July 2023). All stimuli, i.e., videos, are available to view at: https://drive.google.com/drive/folders/19wyub8qJucqt1DcBEi860jIjuAtfqYm5?usp=sharing (accessed on 1 July 2023).

**Conflicts of Interest:** The authors declare no conflict of interests.

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
