# Peer review of "Visual Cognition and the Science of Magic"

_2411-5150, 2023_

Round 1

Reviewer 1 Report

Please see attached file for comments.

Author Response

Please see attached replied to reviewers document. 

Reviewer 2 Report

It has been claimed that the art of conjuring and magic tricks can assist the development of 

theories and knowledge in visual cognition. In a series of seven experiments the paper tested various factors. For example, observers saw a video of a magician performing the French Drop sleight whilst some key elements were hidden: the gaze, motion, or muscular tension. Surprisingly it turns out that these factors are not important. The authors conclude that in the literature there have been a wrong assumption that  magicians have unique insights about human attention and perception.

I enjoyed reading the paper, and I have to say that I was already in agreement with the main thesis. There is an unfortunate tendency by scholars to make exaggerated claims about what professionals "know", and this applies to magicians as much as to artists or to athletes.

I have only some suggestions for improvement:

page 1. " the claim that the field of magic possesses particular knowledge of visual cognition". Here and in other places I was confused by the choice of words. It is a bit odd to claim that a "field" knows something. Are we talking specifically about people who practice magic, or is the point slightly different?

page 3. the section on participants needs more information. I understand it was a study done online, but maybe we can know a bit more. Is it likely that no exclusion criteria were used? When was the data collected? Were participants different in every experiment?

page 4. the statistical analysis is also very minimal, more information is needed.

page 8. the results section of exp 6 only mentions tension and not the opening of the hand, which is the new manipulation, as if this was the same as exp 5. This was confusing.

page 8. please provide pag numbers for the quotes.

Author Response

(The authors gave the same response as above.)

Round 2

Author Response

Thank you again for taking the time to suggest improvements to our ms. As requested, we have now described the rationale for our attempt to avoid a ceiling effect and reduce success of the French Drop. This is placed in the Introduction to Experiment 6. In the Results of that experiment we also acknowledge that having Hand A face-down, as well as Hand B closed, could generate an effect of our manipulation. We have also corrected the typo mentioned.